# Learning Generalizable Risk-Sensitive Policies to Coordinate in Decentralized Multi-Agent General-Sum Games

## Abstract

While various multi-agent reinforcement learning methods have been proposed
in cooperative settings, few works investigate how self-interested learning agents
achieve mutual coordination in decentralized general-sum games and generalize
pre-trained policies to non-cooperative opponents during execution. In this paper,
we present a generalizable and sample efficient algorithm for multi-agent coor-
dination in decentralized general-sum games without any access to other agents'
rewards or observations. Specifically, we first learn the distributions over the return
of individuals and estimate a dynamic risk-seeking bonus to encourage agents to
discover risky coordination strategies. Furthermore, to avoid overfitting opponents'
coordination strategies during training, we propose an auxiliary opponent modeling
task so that agents can infer their opponents' type and dynamically alter corre-
sponding strategies during execution. Empirically, we show that agents trained
via our method can achieve mutual coordination during training and avoid being
exploited by non-cooperative opponents during execution, which outperforms other
baseline methods and reaches the state-of-the-art.

## 1 Introduction

Inspired by advances in deep reinforcement learning (DRL)[1–3], many researchers recently focus
on utilizing DRL methods to tackle multi-agent problems[4–6]. However, most of these works either
consider the fully cooperative multi-agent reinforcement learning (MARL) settings [7–11] or general-
sum games but make restrictive assumptions about opponents[12–14], e.g., either stationary[13]
or altruistic [15, 16]. Considering future applications of MARL, such as self-driving cars[17] and
human-robot interactions [18], multiple learning agents optimize their own rewards independently in
general-sum games where win-win outcomes are only achieved through coordination which often
coupled with risk[19, 12, 20] ("Risk" refers to the uncertainty of future outcomes[21]), and their
pre-trained policies should generalize to non-cooperative opponents during execution.

To achieve coordination alongside other learning agents and generalize learned policies to non-
cooperative opponents, the agent must be willing to undertake a certain amount of risk and identify
the opponents' type efficiently. One set of approaches use explicit reward shaping to force agents to
coordinate[22, 16, 15], which can be viewed as an approach to shape the risk degree of coordination
strategies. To learn generalizable policies, [15, 20] propose to train an adaptive agent with population-
based training methods. Other works either treat the other agents as stationary[13, 23, 24, 20], or
directly access to opponent's policy parameters[12].

Submitted to 36th Conference on Neural Information Processing Systems (NeurIPS 2022). Do not distribute.

By contrast, we are interested in a less restrictive setting where we do not assume access opponents' rewards, observations, or policy parameters, instead, each agent can infer other agents' current strategies from the past behaviors of other agents. In this paper, one key insight is that learning from opponent's past behaviors allows the agent to infer the opponent's type and dynamically alter his strategy between different modes, e.g., either cooperate or compete, during execution. Moreover, given that the other learning agents are non-stationary, decision-making over the agent's return distributions enables the agent to tackle uncertainties resulting from other agents' behaviors and alter his risk preference, i.e., from risk-neutral to risk-seeking, to discover coordination strategies. Motivated by the analysis above, we propose GRSP, a Generalizable Risk-Sensitive MARL algorithm and our contributions are summarized as follows:

**Leading to mutual coordination in decentralized general-sum games.** We estimate a dynamic risk-seeking bonus using a complete distortion risk measure Wang's Transform (WT)[25] to encourage agents to discover risky cooperative strategies. The risk-seeking bonus only affects the action selection procedure instead of shaping environment rewards and decreases throughout training, leading to an unbiased policy.

**Generalizing pre-trained policies to non-cooperative opponents during execution.** Policies learned independently can overfit to the other agents' policies during training, failing to sufficiently generalize during execution[26]. We further propose to train each learning agent with two objectives: a standard Quantile Regression objective[27, 28] and a supervised agent modeling objective, which models the behaviors of opponent, applied on intermediate representation of the value network. The auxiliary opponent modeling task allows the policy to be influenced by opponent's past behaviors, forcing the intermediate representation to adapt to the new opponent.

**Evaluating in multi-agent settings.** We evaluate GRSP in four different Markov games: Monster-Hunt[15, 29], Escalation[15, 16], Iterated Prisoners' Dilemma (IPD)[12, 20] and Iterated Stag Hunt (ISH)[19, 15]. Compared with several baseline methods, including MADDPG[30], MAPPO[31], LIAM[13], IAC[32] and LOLA[12], GRSP learns substantially faster, achieves mutual coordination during training and can generalize to the non-cooperative opponent during execution, which outperforms other baseline methods and reaches the state-of-the-art.

## 2   Related Work

**Risk-sensitive RL.** Risk-sensitive policies, which depend upon more than mean of the outcomes, enable agents to handle the intrinsic uncertainty arising from the stochasticity of the environment. In MARL, the intrinsic uncertainties are amplified due to the non-stationarity and partial observability created by other agents that change their policies during the learning procedure[33–35]. Distributional RL[36, 28] provides a new perspective for optimizing policy under different risk preferences within a unified framework[21, 37]. With distributions of return, it is able to approximate value function under different risk measures, such as Conditional Value at Risk (CVaR)[38, 39] and WT[25], and thus produce risk-averse or risk-seeking policies. Qiu et al.[11] propose RMIX with the CVaR measure as risk-averse policies. Similar ideas are proposed in D4PG[40] and DFAC[41]. In contrast with these works that focus on the fully cooperative settings and do not consider generalization, this paper proposes the first algorithm that leverages risk-seeking policies to achieve coordination strategies in general-sum games and generalizable to non-cooperative opponents during testing phase.

**Generalization across different opponents.** Many real world scenarios require agents to adapt to different opponents during execution. However, most of existing works focus on learning a fixed and team-dependent policy in fully cooperative setting[42, 8, 9, 11, 10] which can not generalize to slightly altered environments or new opponents. Other works either use a population-based training method to train an adaptive agent[15], or adapt to different opponents under the Tit-for-Tat principle[20, 43]. Our work is closely related to test-time training methods[44, 45]. However, they focus on image recognition or single agent policy adaption. Ad hoc teamwork[46, 47] also requires agents to generalize to new teams, but they focus on cooperative games and has different concerns with us.

83 **Opponent modeling.** Our approach to learning generalizable policies can be viewed as a kind
84 of opponent modeling method[48]. These approaches either model intention[49, 50], assume an
85 assignment of roles[51] or exploit opponent learning dynamics[12, 52]. Our approach is similar
86 to policy reconstruction methods[50] which make explicit predictions about opponent's actions.
87 However, instead of predicting the opponent's future actions, we learn from opponent's past behaviors
88 to update the belief, i.e., parameters of value network, of the opponent's type.

## 89 3 Preliminaries

90 **Stochastic games.** In this work, we consider multiple self-interested learning agents interact with
91 each other. We model the problem as a Partially-Observable Stochastic Game (POSG)[53, 54], which
92 consists of $N$ agents, a state space $\mathcal{S}$ describing the possible configurations of all agents, a set of
93 actions $\mathcal{A}^1, \ldots, \mathcal{A}^N$ and a set of observations $\mathcal{O}^1, \ldots, \mathcal{O}^N$ for each agent. At each time step, each
94 agent $i$ receives his own observation $o^i \in \mathcal{O}^i$, and selects an action $a^i \in \mathcal{A}^i$ based on a stochastic
95 policy $\pi^i : \mathcal{O}^i \times \mathcal{A}^i \mapsto [0, 1]$, which results in a joint action vector $\boldsymbol{a}$. The environment then
96 transitions to a new state $s'$ based on the transition function $P(s'|s, \boldsymbol{a})$. Each agent $i$ obtains rewards
97 as a function of the state and his action $R^i : \mathcal{S} \times \mathcal{A}^i \mapsto \mathbb{R}$. The initial states are determined by a
98 distribution $\rho : \mathcal{S} \mapsto [0, 1]$. We treat the reward "function" $R^i$ of each agent as a random variable to
99 emphasize its stochasticity, and use $Z^{\pi^i}(s, a^i) = \sum_{t=0}^{T} \gamma^t R^i(s_t, a_t^i)$ to denote the random variable
100 of the cumulative discounted rewards where $S_0 = s, A_0^i = a^i, \gamma$ is a discount factor and $T$ is the time
101 horizon.

102 **Distorted expectation.** Distorted expectation is a risk weighted expectation of value distribution
103 under a specific distortion function[55]. A function $g : [0, 1] \mapsto [0, 1]$ is a distortion function if it is
104 non-decreasing and satisfies $g(0) = 0$ and $g(1) = 1$[56]. The distorted expectation of $Z$ under $g$ is
105 defined as $\Psi(Z) = \int_0^1 F_Z^{-1}(\tau) dg(\tau) = \int_0^1 g'(\tau) F_Z^{-1}(\tau) d\tau$, where $F_Z^{-1}$ is the quantile function at
106 $\tau \in [0, 1]$ for the random variable $Z$. We introduce two common distortion functions as follow:

107 • **CVaR** is the expectation of the lower or upper tail of the value distribution, corresponding to
108    risk-averse or risk-seeking policy respectively. Its distortion function is $g(\tau) = \min(\tau/\alpha, 1)$
109    (risk-averse) or $\max(0, 1 - (1 - \tau)/\alpha)$ (risk-seeking), $\alpha \in (0, 1)$ denotes confidence level.

110 • **WT** is proposed by Wang[25]: $g_\lambda(\tau) = \Phi\left(\Phi^{-1}(\tau) + \lambda\right)$, where $\Phi$ is the distribution of a standard
111    normal. The parameter $\lambda$ is called the market price of risk and reflects systematic risk. $\lambda > 0$ for
112    risk-averse and $\lambda < 0$ for risk-seeking.

113 $\text{CVaR}_\alpha$ assigns a 0-value to all percentiles below the $\alpha$ or above $1 - \alpha$ significance level which leads
114 to erroneous decisions in some cases[56]. Instead, WT is a complete distortion risk measure and
115 ensures using all the information in the original loss distribution which makes training much more
116 stable, and we will empirically demonstrate it in Sec. 5.

## 117 4 Methods

118 In this section, we describe our proposed GRSP method. We first introduce the risk-seeking bonus
119 used to encourage agents to discover coordination strategies in Sec. 4.1 and then propose the auxiliary
120 opponent modeling task to learn generalizable policies in Sec. 4.2. Finally, we provide the details of
121 test-time policy adaptation under different opponents in Sec. 4.3.

### 122 4.1 Risk-Seeking Bonus

123 In this section, we first provide an illustrative example for the insight behind risk-seeking bonus and
124 then describe its details. Consider a two-player 10 steps Sequential matrix game Stag Hunt, where
125 each player should decide whether to hunt stag (S) or hunt hare (H) in each round. If both agents
126 choose S they will receive the highest payoff 2. However, if one agent defects, he will receive a

127 descent reward 1 for eating the hare alone while the other agent with an S action will suffer from a
128 big loss -10. If both agents choose H they will receive payoff 1.

129 Even state of the art RL algorithms fail to discover the
130 "risky" cooperation strategies[15, 16, 19]. One important
131 reason is that the expected, i.e., risk-neutral, Q value ig-
132 nores the complete distribution information, especially the
133 upper and lower tail information when the learned distribu-
134 tion is asymmetric. Another reason is that when the risk is
135 high, i.e., a high loss for being betrayed, the probability of
136 finding the S-S (Cooperation) strategy via policy gradient
137 is very low[15].

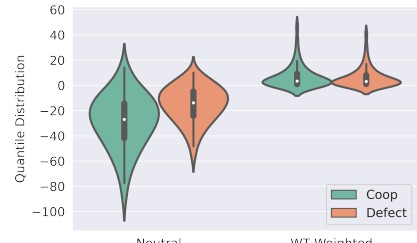

138 Therefore, we adopt the distributional RL method to model
139 the whole distribution of Q value. Fig.1 left part shows
140 the quantile distribution of cooperation and defection of
141 risk-neutral policy learned by QR-DQN[28]. The mean
142 value of defection is higher than that of cooperation, but the quantile value distribution of cooperation
143 has a longer upper tail which means that it has a higher potential payoff.

Figure 1: Quantile value distribution of cooperation and defection in Sequential Stag Hunt weighted by WT compared with risk-neutral policy.

144 We propose to use WT distortion function to reweight the expectation of quantile distribution. By
145 following [28], we first represent the return distribution of each agent $i$ with policy $\pi^i$ by a uniform
146 mix of $M$ supporting quantiles:

$$Z_\theta^{\pi^i}(o^i, a^i) \doteq \frac{1}{M} \sum_{k=1}^{M} \delta_{\theta_k^{\pi^i}(o^i, a^i)} \tag{1}$$

147 where $\delta_x$ denotes a Dirac Delta functions at $x \in \mathbb{R}$, and each $\theta_k^{\pi^i}$ is an estimation of the quantile
148 corresponding to the quantile fractions $\hat{\tau}_k \doteq \frac{\tau_{k-1} + \tau_k}{2}$ with $\tau_k \doteq \frac{k}{M}$ for $0 \le k \le M$. The state-action
149 value $Q^{\pi^i}(o^i, a^i)$ can then be approximated by $\frac{1}{M} \sum_{k=1}^{M} \theta_k^{\pi^i}(o^i, a^i)$.

150 Furthermore, the risk-seeking bonus for agent $i$ is defined as:

$$\Psi(Z_\theta^{\pi^i}) = \int_0^1 g_\lambda'(\tau) F_{Z_\theta^{\pi^i}}^{-1}(\tau) d\tau \approx \frac{1}{M} \sum_{k=1}^{M} g_\lambda'(\hat{\tau}_k) \theta_k^i, \tag{2}$$

151 where $g_\lambda'(\tau)$ is the derivatives of WT distortion function at $\tau \in [0, 1]$, and $\lambda$ controls the risk-seeking
152 level. Fig.1 right part shows the WT weighted quantile distribution in which the upper quantile values
153 are multiplied by bigger weights and lower quantile values are multiplied by smaller weights to
154 encourage agents to adopt risky coordination strategies.

155 A naive approach to exploration would be to use the variance of the estimated distribution as a bonus.
156 [57] shows that the exploration bonus from truncated variance outperforms bonus from the variance.
157 Specifically, the Right Truncated Variance tells about lower tail variability and the Left Truncated
158 Variance tells about upper tail variability. For instantiating optimism in the face of uncertainty, the
159 upper tail variability is more relevant than the lower tail, especially if the estimated distribution is
160 asymmetric. So we adopt the Left Truncated Variance of quantile distribution to further leverage the
161 intrinsic uncertainty for efficient exploration. The left truncated variance is defined as

$$\sigma_+^2 = \frac{1}{2M} \sum_{j=\frac{M}{2}}^{M} \left( \theta_{\frac{M}{2}} - \theta_j \right)^2, \tag{3}$$

162 and analysed in [57]. The index starts from the median, i.e., $M/2$, rather than the mean due to its well-
163 known statistical robustness[58, 59]. We anneal the two exploration bonuses dynamically so that in
164 the end we produce unbiased policies. The anneal coefficients are defined as $c_{tj} = c_j \sqrt{\frac{\log t}{t}}, j = 1, 2$
165 which is the parametric uncertainty decay rate[60], and $c_j$ is a constant factor. This approach leads to

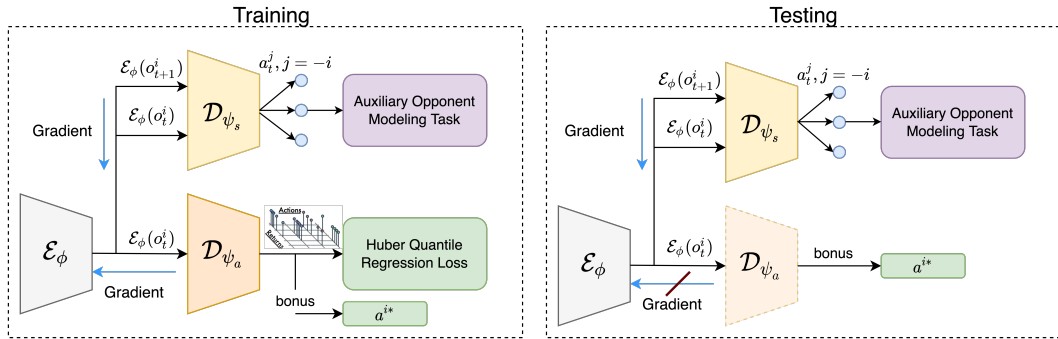

Figure 2: **Left:** Diagram of GRSP architecture during training. Outputs of $\mathcal{E}_\phi$ are fed into $\mathcal{D}_{\psi_a}$ and $\mathcal{D}_{\psi_s}$, so features are shared between policy and auxiliary opponent modeling. The prediction head $\mathcal{D}_{\psi_s}$ outputs other agents' actions. **Right:** Test-Time policy adaptation. The agent can not receive environment rewards during testing, so we only optimize the auxiliary opponent modeling objective.

choosing the action such that

$$a^{i*} = \arg\max_{a^i \in \mathcal{A}^i} \left( Q^{\pi^i}(o^i, a^i) + c_{t1}\Psi(Z^{\pi^i}(o^i, a^i)) + c_{t2}\sqrt{\sigma^2_+(o^i, a^i)} \right) \tag{4}$$

These quantile estimates are trained using the Huber[61] quantile regression loss. The loss of the quantile value network of each agent $i$ at time step $t$ is then given by

$$\mathcal{J}\left(o^i_t, a^i_t, r^i_t, o^i_{t+1}; \theta^i\right) = \frac{1}{M}\sum_{k=0}^{M-1}\sum_{j=0}^{M-1}\rho^\kappa_{\hat{\tau}_k}\left(\delta^{ti}_{kj}\right) \tag{5}$$

where $\delta^{ti}_{kj} \doteq r^i_t + \gamma\theta^i_j\left(o^i_{t+1}, \pi^i\left(o^i_{t+1}\right)\right) - \theta^i_k(o^i_t, a^i_t)$, and $\rho^\kappa_{\hat{\tau}_k}(x) \doteq |\hat{\tau}_k - \mathbb{I}\{x < 0\}|\frac{\mathcal{L}_\kappa(x)}{\kappa}$ where $\mathbb{I}$ is the indicator function and $\mathcal{L}_\kappa(x)$ is the Huber loss:

$$\mathcal{L}_\kappa(x) \doteq \begin{cases} \frac{1}{2}x^2 & \text{if } x \leq \kappa \\ \kappa\left(|x| - \frac{1}{2}\kappa\right) & \text{otherwise} \end{cases} \tag{6}$$

## 4.2 Auxiliary Opponent Modeling Task

In order to alter the agent's strategies under different opponents, we share parameters between policy and auxiliary opponent modeling task. Specifically, we split the $Q$ value network into two parts: feature extractor $\mathcal{E}_\phi$ and decision maker $\mathcal{D}_{\psi_a}$. The parameters of the $Q$ value network $Q_{\theta^i}$ for agent $i$ are sequentially divided into $\phi^i$ and $\psi^i_a$, i.e., $\theta^i = (\phi^i, \psi^i_a)$. The auxiliary opponent modeling task shares a common feature extractor $\mathcal{E}_{\phi^i}$ with the value network. We can update the parameters of $\mathcal{E}_{\phi^i}$ during execution using gradients from the auxiliary opponent modeling task, such that $\pi_{\theta^i}$ can generalize to different opponents. The supervised prediction head and its specific parameters are $\mathcal{D}_{\psi^i_s}$ with $\psi^i_s$. The details of our network architecture are shown in Fig. 2.

During training, the agent $i$ can collect a set of transitions $\{(o^i_t, o^i_{t+1}, \mathbf{a}^{-i}_t)\}^T_{t=0}$ where $\mathbf{a}^{-i}_t$ indicates the joint actions of other agents except $i$ at time step $t$. We use the embeddings of agent $i$'s observations $o^i_t$ and $o^i_{t+1}$ to predict the joint actions $\mathbf{a}^{-i}_t$, i.e., the $\mathcal{D}_{\psi^i_s}$ is a multi-head neural network whose outputs are multiple soft-max distributions over the discrete action space or predicted continuous actions of each other agent, and the objective function of the auxiliary opponent modeling task can be formulated as

$$\mathcal{L}\left(o^i_t, o^i_{t+1}, \mathbf{a}^{-i}_t; \phi^i, \psi^i_s\right) = \frac{1}{N-1}\sum_{j=1, j\neq i}^N \ell\left(a^j_t, \mathcal{D}_{\psi^i_s}\left(\mathcal{E}_{\phi^i}\left(o^i_t\right), \mathcal{E}_{\phi^i}\left(o^i_{t+1}\right)\right)^j\right), \tag{7}$$

where $\ell(\cdot)$ is the cross-entropy loss function for discrete actions or mean squared error for continuous actions. The strategies of opponents will change constantly during the procedure of multi-agent exploration and thus various strategies will emerge. The agent can leverage them to gain some experience about how to make the best response by jointly optimizing the auxiliary opponent modeling task and quantile value distribution. The joint training problem is therefore

$$\min_{\phi^i, \psi^i_s, \psi^i_a} \mathcal{J}\left(o^i_t, a^i_t, r^i_t, o^i_{t+1}; \phi^i, \psi^i_a\right) + \mathcal{L}\left(o^i_t, o^i_{t+1}, \mathbf{a}^{-i}_t; \phi^i, \psi^i_s\right) \tag{8}$$

### 4.3 Test-Time Policy Adaptation under Different Opponents

During testing time, we can not optimize $\mathcal{J}$ anymore since the reward is unavailable, but we assume the agent can observe actions made by his opponents during execution, then we can continue optimizing $\mathcal{J}$ to update the parameters of feature extractor $\mathcal{E}_\phi$. Learning from opponents' past behaviors at test time makes the agent generalize his policy to different opponents efficiently. The can be formulated as

$$\min_{\phi^i, \psi^i_s} \mathcal{L}\left(o^i_t, o^i_{t+1}, \mathbf{a}^{-i}_t; \phi^i, \psi^i_s\right) \tag{9}$$

## 5 Experiments

In this section, we empirically evaluate our method on four multi-agent environments. In sec. 5.1 we introduce the four environments we use for experiments and training settings. In sec. 5.2 we compare the performance of GRSP with other baselines. In sec. 5.3 we evaluate the generalization ability of GRSP under different opponents during execution. The ablations are studied in sec. 5.4. Further understanding of GRSP is presented in sec. 5.5. More details can be found in Appendix C.

### 5.1 Environment Setup

**Repeated games.** We consider two kinds of repeated matrix games: Iterated Stag Hunt (ISH) and Iterated Prisoners' Dilemma (IPD). Both of them consist two agents and a constant episode length of 10 time steps[12, 15, 19]. At each time step, the agents can choose either cooperation or defection. If both agents choose to cooperate simultaneously, they both get a bonus of 2. However, if a single agent choose to cooperate, he gets a penalty of -10 in ISH and -1 in IPD, and the other agent get a bonus of 1 and 3, respectively. If both agents choose to defection, they get a bonus of 1 in ISH and 0 in IPD. The optimal strategy in ISH and IPD is to cooperate at each time step, and the highest global payoffs of two agents are 40, i.e., 20 for each of them.

**Monster-Hunt.** The environment is a $5 \times 5$ grid-world, consisting of two agents, two apples and one monster. The apples are static while the monster keeps moving towards its closest agent. When a single agent meets the monster, he gets a penalty of -10. If two agents catch the monster together, they both get a bonus of 5. If a single agent meets an apple, he get a bonus of 2. Whenever an apple is eaten or the monster meets an agent, the entity will respawn randomly. The optimal strategy, i.e., both agents move towards and catch the monster, is a risky coordination strategy since an agent will receive a penalty if the other agent deceives.

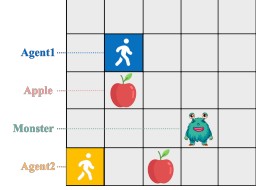

Figure 3: Monster-Hunt.

**Escalation.** Escalation is a $5 \times 5$ grid-world with sparse rewards, consisting of two agents and a static light. If both agents step on the light simultaneously, they receive a bonus of 1, and then the light moves to a random adjacent grid. If only one agent steps on the light, he gets a penalty of $1.5L$, where $L$ denotes the latest consecutive cooperation steps, and the light will respawn randomly. To maximize their individual payoffs and global rewards, agents must coordinate to stay together and step on the light grid. For each integer L, there is a corresponding

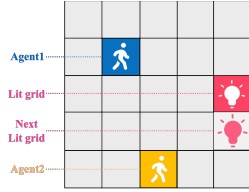

Figure 4: Escalation.

coordination strategy where each agent follows the light for $L$ steps then simultaneously stop coordination.

**Training.** We carry out our experiments on one NVIDIA RTX 3080 Ti and Intel i9-11900K.

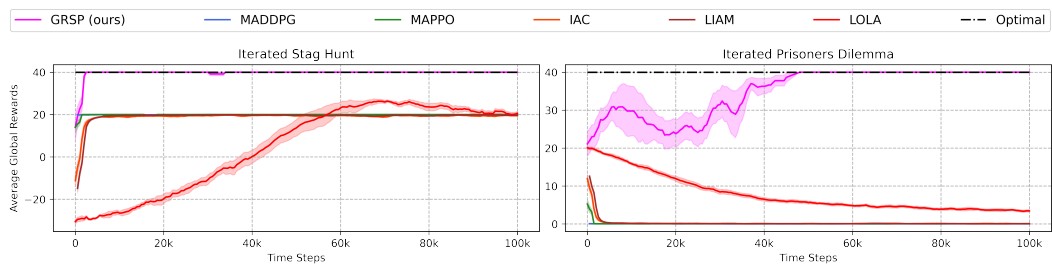

Figure 5: Mean evaluation returns for GRSP, MADDPG, MAPPO, IAC, LIAM and LOLA on two repeated matrix games. The average global rewards equal to 40 means that all agents have learned coordination strategy, i.e., cooperating at each time step.

## 5.2    Evaluation of Returns

In this subsection, we evaluate all methods on four multi-agent environments and use 5 different random seeds to train each method. We pause training every 50 episodes and run 30 independent episodes with each agent performing greedy action selection to evaluate the average performance of each method.

### 5.2.1    Iterated Games

Fig. 5 shows the average global rewards, i.e., the summation of all agents' average returns, of all methods evaluated during training in ISH and IPD environments. The shadowed part represents a 95% confidence interval. The average global rewards equal to 40 means that all agents have learned coordination strategy, i.e., cooperating at each time step. We can find that agents trained with our method can achieve mutual coordination in a sample efficient way in two repeated matrix games with high risk while other methods only converge to safe non-cooperative strategies though some of them have much more restrictive assumptions.

### 5.2.2    Grid-Worlds

We further show the effectiveness of GRSP in two grid-world games, Monster-Hunt and Escalation[15], both of which have high payoff but risky cooperation strategies for agents to converge to. Fig. 6. shows that, compared with other baseline methods, GRSP constantly and significantly outperform baselines with higher sample efficiency over the whole training process both in global rewards and agent's individual rewards. Specifically, in Monster-Hunt, GRSP agents efficiently find one of the risky cooperation strategies where two agents stay together and wait for the monster. Furthermore, the policies learned by each agent are very stable and neither would like to deviate from the cooperative strategy. However, other baseline methods only converge to safe non-cooperative strategies and get low payoff due to their poor exploration. It seems that LOLA can not learn useful strategies in more complex environments. In Escalation, GRSP outperforms other baselines significantly and both agents have achieved coordination in a decentralized paradigm.

## 5.3    Generalization Study

This subsection investigates how well the pre-trained GRSP agent can generalize to different opponents, i.e., cooperation or defection, during execution. The cooperative opponents are trained by GRSP method while the non-cooperative opponents are trained by MADDPG. During evaluation, random seeds of four environments are different from that during training, and hyperparameters

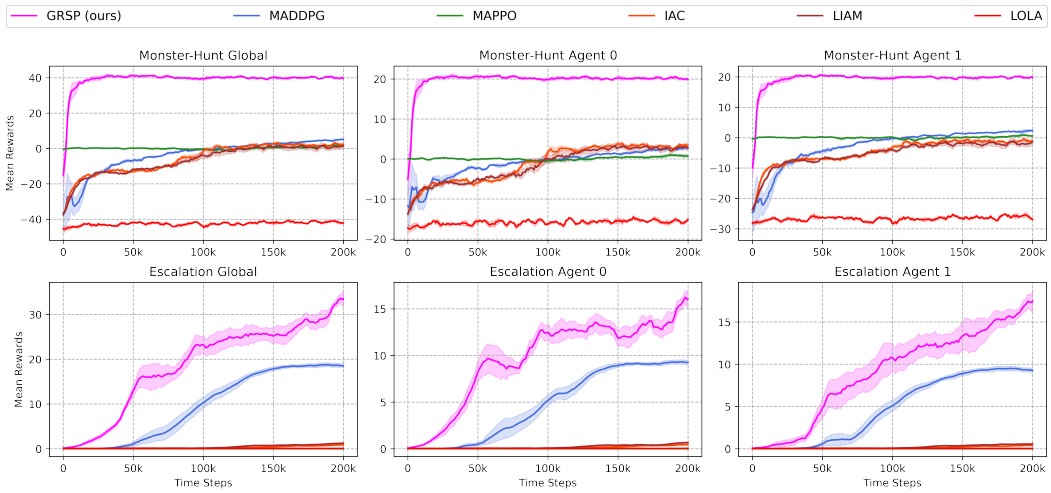

Figure 6: Mean evaluation returns for GRSP, MADDPG, MAPPO, IAC, LIAM and LOLA on Monster-Hunt and Escalation. Global rewards are summation of both agents rewards.

Table 1: Mean evaluation return of GRSP with and without auxiliary opponent modeling task on four multi-agent encironments.

| Oppo: Coop(Defect) | ISH | IPD | M-H | Escalation |
|---|---|---|---|---|
| GRSP-No-Aom | $20(-100)$ | $20(-5)$ | $20.62(-15.03)$ | $9.45(-0.545)$ |
| GRSP-Aom | $\mathbf{20(0.65)}$ | $\mathbf{20(-1.08)}$ | $\mathbf{21.36(-12.07)}$ | $\mathbf{11.3(0.175)}$ |

of the GRSP are same and fixed between different opponent types. Furthermore, the pre-trained coordinated agents can not access to the rewards to update their policies anymore and they must utilize the auxiliary opponent modeling task to force them to adapt to different opponents. The network details and hyperparameters can be found in Appendix B.

Table 1 shows the mean evaluation return of GRSP agent with and without the auxiliary opponent modeling task on four multi-agent environments when interacting with different opponents. All returns are averaged on 100 episodes. The performance of the GRSP-Aom agent that utilizes the auxiliary opponent modeling task to adapt to different opponents outperforms that of the GRSP-No-Aom agent significantly, especially when interacting with non-cooperative opponents. Specifically, the GRSP-Aom agent using history behaviors of its opponents to update its policy can learn to alter its strategy from coordination to not when encountering a non-cooperative opponent. The empirical results further demonstrate that policies learned independently can overfit to the other agents' policies during training, and our auxiliary opponent modeling task provides a method to tackle this problem.

## 5.4 Ablations

In this subsection, we perform an ablation study to examine the components of GRSP to better understand our method. GRSP is based on QR-DQN and has three components: the risk-seeking exploration bonus, the left truncated variance (Tv) and the auxiliary opponent modeling task (Aom). We design and evaluate six different ablations of GRSP in two grid-world environments, as show in Fig. 7. The performance of GRSP-No-Aom which we ablate the Aom module and retain all other features of our method is a little lower than that of GRSP but has a much higher variance, indicating that learning from opponent's behaviors can stable training and improve performance. Moreover, the GRSP-No-Aom is a completely decentralized method whose training without any opponent information, and the ablation results of GRSP-No-Aom show that our risk-seeking bonus is essential for agents to achieve mutual coordination in general-sum games. We observe that ablating the left truncated variance module leads to a significantly lower return than the GRSP in the Escalation

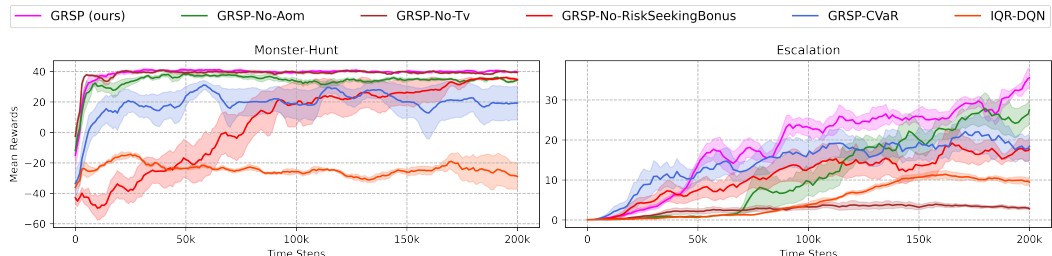

Figure 7: Mean evaluation return of GRSP compared with other ablation methods in two grid-world multi-agent environments.

but no difference in the Monster-Hunt. Furthermore, ablating the risk-seeking bonus increases the training variance, leads to slower convergence and perform worse than the GRSP. It is noteworthy that the Escalation is a sparse reward and hard-exploration multi-agent environment since our two decentralized agents can get a reward only if they navigate to and step on the light simultaneously and constantly. These two ablations indicate that the exploration ability of left truncated variance is important to our method and the risk-seeking bonus can encourage agents to coordinate with each other stably and converge to high-risky cooperation strategies efficiently. We also implement our risk-seeking bonus by CVaR instead of WT, and the results are shown as GRSP-CVaR. The GRSP-CVaR performs worse than our method and has a higher training variance. Finally, we ablate all components of the GRSP and use $\epsilon$-greedy policy for exploration which leads to the IQR-DQN algorithm. As shown in Fig. 7, IQR-DQN can not learn effective policies in the Monster-Hunt and perform badly in the Escalation.

## 5.5 Understanding GRSP

The action whose value distribution has a long upper tail means that taking this action may receive higher potential payoffs. However, its mean value may be lower than other actions since its distribution has a longer lower tail, as shown in Fig. 1 Neutral-Coop, indicating higher risk. So agents with the expected RL method will not select this action. In GRSP, the risk-seeking exploration bonus encourages agents to pay more attention to actions whose distribution has a longer upper tail. So agents with GRSP method will be less likely to defect their opponents since defects bring lower future returns, more likely to coordinate with other agents, and more tolerant of the risk. Furthermore, the auxiliary opponent modeling task can alter the agent's strategy from cooperation to defection if it pairs with a non-cooperative opponent. Empirically, the two components can constitute a kind of equilibrium strategies, e.g., tit-for-tat[20], between agents.

## 6 Discussion

**Conclusion.** While various MARL methods have been proposed in cooperative settings, few works investigate how self-interested learning agents can achieve mutual coordination which is coupled with risk in decentralized general-sum games and generalize learned policies to non-cooperative opponents during execution. In this paper, we present GRSP, a novel decentralized MARL algorithm with estimated risk-seeking bonus and auxiliary opponent modeling task. Empirically, we show that agents trained via GRSP can not only achieve mutual coordination during training with high sample efficiency but generalize learned policies to non-cooperative opponents during execution, while other baseline methods can not.

**Limitations and future work.** The risk-seeking bonus in GRSP is estimated using WT distorted expectation and its risk-sensitive level is a hyperparameter that can not dynamically change throughout training. Developing a method that can adjust agents' risk-sensitive levels dynamically by utilizing their observation, rewards, or opponents' information is the direction of our future work.

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
