# OpenReview forum: "Learning Generalizable Risk-Sensitive Policies to Coordinate in Decentralized Multi-Agent General-Sum Games"
_NeurIPS.cc/2022/Conference — NeurIPS 2022 Submitted_

### Official Review · Reviewer_DkS2 · 2022-06-30

**Rating:** 4
**Confidence:** 4
**Soundness:** 2 fair
**Presentation:** 3 good
**Contribution:** 2 fair

**Summary:**

This paper suggests learning risk-seeking policies in general-sum games in order to promote cooperation. The authors seek the strategy using distributional RL and risk-seeking weighting. Empirical evaluation on a range of domains shows that this algorithm finds the cooperative outcomes where other multi-agent RL methods do not, even though some of them focus on opponent modeling and coordination.

**Questions:**

Why did you not compare to the existing algorithms for learning cooperative outcomes in general sum games?

What are the main reasons why the proposed algorithm finds the cooperation and the existing ones fain in some of the specific domains?

**Limitations:**

There is not obvious potential negative societal impact. The limitations are IMO not explored sufficiently well. It is not clear when the proposed algorithm starts failing.

**Strengths And Weaknesses:**

*Positives*

I like the idea of using risk-seeking behavior for creating coordination and I like the use of distributional RL for implementing it. I also like the empirical evaluation, which uses a sufficiently wide range of domains and clearly shows the superiority of the proposed approach over the chosen baselines.

*Major concerns*

My main concerns are lack of theoretical grounding and depth of understanding conveyed by the experimental results.

Learning in games is a relatively old and well studied field with many solution concepts and algorithms to achieve them. This paper presents an intuitive solution to promote cooperation, but does not analyze at all what the proposed algorithm is supposed to converge to theoretically. Are there any guarantees on the behavior of the algorithm even in simple matrix games? Is there some relation between the parameters for risk seeking and the gap between cooperative and non-cooperative equilibria of the game? Does the algorithm converge to correlated equilibria as many other learning dynamics under some conditions or even that is not guaranteed?

I believe that theoretical understanding of algorithms is important, but not absolutely necessary for a good paper, if the empirical evaluation chooses the right baselines and provides an in-depth analysis of why and when the proposed algorithm is superior. Since there are many papers dealing with trying to learn cooperative outcomes in general sum games (such as [A,B]) I consider the choice of baselines insufficient. Furthermore, the experiments do not provide any insights into when and why the proposed method performs better. The ablation study is definitely a step in the right direction, but I still do not know when to expect the algorithm to work well and when not.

[A] Littman ML. Friend-or-foe Q-learning in general-sum games. InICML 2001 Jun 28 (Vol. 1, pp. 322-328).
[B] Crandall JW, Goodrich MA. Learning to compete, compromise, and cooperate in repeated general-sum games. InProceedings of the 22nd international conference on Machine learning 2005 Aug 7 (pp. 161-168).

To summarize, the overall idea of the paper is interesting to me, but the thoroughness of setting in the related work and the depth of the analysis is slightly below the bar I would expect from a NeurIPS publication.

*Minor suggestions*

Lines 31-32 claim that training a population of strategy profiles is infeasible in real world settings, which does not seem to be true with algorithms, such as AplhaStar.

It is not clear what exactly are the error areas in the graphs. Confidence intervals? Variance? STD?

Some formulations are hard to follow, such as L196-197, L236,

---

> ### Author Response · Authors · 2022-08-02
> **Responses to Reviewer DkS2**
>
> We thank the reviewer’s helpful feedback on our submission. We summarize the reviewer’s questions and present our responses below.
>
> Q1: the experiments do not provide any insights into when and why the proposed method performs better.
>
> A1: We have revised the experiment part of our paper and updated them in the rebuttal revision. You can find our insights in sec1, sec. 4.1, sec 5.4, and sec. 5.5. And we are willing to keep revising our paper according to your valuable further feedback.
> ***
>
> Q2: It is not clear what exactly are the error areas in the graphs. Confidence intervals? Variance? STD?
>
> A2: The shadowed part represents a 95% conﬁdence interval, i.e., two standard errors of the mean. We
> ***
> Q3: Some formulations are hard to follow, such as L196-197, L236,
>
> A3: We have revised the method and experiment part of our paper and updated them in the rebuttal revision.
> ***
> Q4: Why did you not compare to the existing algorithms for learning cooperative outcomes in general sum games?
>
> A4: To our knowledge, there are few works studying multi-agent coordination in general-sum games since 2016. Most related works that study general-sum games assume their opponents have fixed policies, e.g., [B],[C],[D]. However, in our work, we consider achieving coordination in multiple learning agents, which is a more general case. We have compared our GRSP with [C] because it is the latest work. The problems studied in [A], [E], [F], and [G] are more related to ours. [A] proposed that agents can exchange their parameters and gradients with their opponents, which is a too strong assumption. We have compared our GRSP with [A] LOLA in the rebuttal revision. The method proposed in [E] is too weak and can not achieve complete cooperation even in iterated stag hunt games. [F] and [G] belong to reward shaping methods, and we have compared GRSP with [F] in the appendix of our paper. The method proposed in [G] can not match their official codes, so we didn't compare GRSP with them.
>
> The reason why we compare GRSP with IAC, MADDPG, and MAPPPO is that the assumptions of these methods are similar to ours, e.g., multiple learning agents, can not access opponent's rewards or parameters, and so on. However, MADDPG assumes access to the opponent's observations and actions to train a centralized critic, which is stronger than ours. But our GRSP method still has better performance than theirs. We think this is one of the contributions of our method.
>
> [A] Foerster, Jakob N., et al. "Learning with opponent-learning awareness." arXiv preprint arXiv:1709.04326 (2017).
>
> [B] Raileanu, Roberta, et al. "Modeling others using oneself in multi-agent reinforcement learning." International conference on machine learning. PMLR, 2018.
>
> [C] Papoudakis, Georgios, Filippos Christianos, and Stefano Albrecht. "Agent modelling under partial observability for deep reinforcement learning." Advances in Neural Information Processing Systems 34 (2021): 19210-19222.
>
> [D] Wang, Weixun, et al. "Towards cooperation in sequential prisoner's dilemmas: a deep multiagent reinforcement learning approach." arXiv preprint arXiv:1803.00162 (2018).
>
> [E] Wang, Woodrow Z., et al. "Emergent prosociality in multi-agent games through gifting." arXiv preprint arXiv:2105.06593 (2021).
>
> [F] Peysakhovich, Alexander, and Adam Lerer. "Prosocial learning agents solve generalized stag hunts better than selfish ones." arXiv preprint arXiv:1709.02865 (2017).
>
> [G] Tang, Zhenggang, et al. "Discovering diverse multi-agent strategic behavior via reward randomization." arXiv preprint arXiv:2103.04564 (2021).
> ***
> Q5: What are the main reasons why the proposed algorithm finds the cooperation and the existing ones fain in some of the specific domains?
>
> A5:
> The action whose value distribution has a long upper tail means that taking this action may receive higher potential payoffs. However, its mean value may be lower than other actions since its distribution also has a longer lower tail, which means higher risk. Agents with the expected RL method (risk-neutral policy) will not select this action, so they can't converge to the cooperation strategies.
>
> In GRSP, the risk-seeking exploration bonus encourages agents to care more about actions whose distribution has a longer upper tail. So agents with the GRSP method will be less likely to defect to their opponents since defects bring lower future returns, more likely to coordinate with other agents, and more tolerant of the risk.

---

> > ### Comment · Reviewer_DkS2 · 2022-08-05
> > **No change in opinion.**
> >
> > The authors did not explain why they do not compare to the pre-existing literature. There are decades of literature on the problem they study and I believe the paper needs to relate to this literature to be acceptable.
> >
> > I do not understand what the authors mean that by "few works studying multi-agent coordination in general-sum games since 2016". How it that relevant? There are many works before that year that the paper IMO needs to relate to.

---

> > > ### Author Response · Authors · 2022-08-06
> > > **Responses to Reviewer DkS2**
> > >
> > > We thank the reviewer’s feedback. We have explained the reasons why we do not compare to the pre-existing literature in A4. We have added a new experiment that compares our method with LOLA [1] and you can find it in the rebuttal revision.  According to your suggestion, we will further supplement the comparison experiments with [2] and [3] as soon as possible.
> > > ***
> > > [1]Foerster, Jakob N., et al. "Learning with opponent-learning awareness." arXiv preprint arXiv:1709.04326 (2017).
> > >
> > > [2]Learning to compete, compromise, and cooperate in repeated general-sum games. InProceedings of the 22nd international conference on Machine learning.
> > >
> > > [3]Littman ML. Friend-or-foe Q-learning in general-sum games. In ICML 2001 Jun 28 (Vol. 1, pp. 322-328).

---

> > > ### Author Response · Authors · 2022-08-06
> > > **Responses to Reviewer DkS2**
> > >
> > > Dear reviewer DkS2, we have supplemented the comparison experiments with [1] and [2]. The mean evaluation rewards of the last 30k time steps of all methods are shown in the table below, and the complete experiment results and our codes can be found at https://anonymous.4open.science/r/GRSP-8DEC/pic/GW.png
> > >
> > > We are looking forward to your further feedback, and we will appreciate it if you can kindly consider raising the score. Thank you.
> > >
> > >
> > > |         | ISH    | IPD    | M-H     | Escalation |
> > > |:-------:|:------:|:------:|:-------:|:----------:|
> > > | GRSP    | **$\mathbf{40.00 \pm 0.0}$**  | **$\mathbf{40.00\pm 0.0}$**  | **$\mathbf{40.04\pm 0.52}$**   | **$\mathbf{30.17\pm 1.39}$**      |
> > > | M-Qubed | $18.67\pm 0.12$  | $10.12\pm 0.29$  | $2.62\pm 0.36$    | $17.42\pm 0.08$      |
> > > | LOLA    | $23.04\pm 1.22$  | $3.92\pm 0.32$   | $41.61\pm 0.49$ | $0.06\pm 0.01$      |
> > > | MADDPG  | $19.91\pm 0.03$  | $0.02\pm 0.01$  | $4.41\pm 0.32$    | $18.62\pm 0.44$      |
> > > | MAPPO   | $20.00\pm0.02$  | $0.05\pm0.01$  | $1.17\pm 0.44$    | $0.00\pm 0.00$       |
> > > | IAC     | $19.41\pm0.13$  |$0.09\pm0.02$  | $1.90\pm 1.57$    | $0.73\pm 0.10$       |
> > > | LIAM    | $19.69\pm 0.10$  | $0.05\pm 0.02$   | $0.77\pm 0.57$    | $1.04\pm 0.09$       |
> > >
> > > ***
> > > [1] Crandall, Jacob W., and Michael A. Goodrich. "Learning to compete, compromise, and cooperate in repeated general-sum games." Proceedings of the 22nd international conference on Machine learning. 2005.
> > >
> > > [2] Foerster, Jakob N., et al. "Learning with opponent-learning awareness." arXiv preprint arXiv:1709.04326 (2017).

---

> > > ### Author Response · Authors · 2022-08-09
> > > **Dear Reviewer DkS2, did our responses address your questions?**
> > >
> > > Dear Reviewer DkS2,
> > >
> > > We have supplemented the comparison experiments with pre-existing literature, and our responses with reviewer ZHUo may eliminate some of your confusion. As the response system will be closed soon within one day. We thank you again for your comments. We hope our detailed responses could address your questions. If there are no more questions and we will appreciate it if you can kindly raise the score.
> > >
> > > Sincerely yours,
> > >
> > > Authors of Paper2297

---

### Official Review · Reviewer_metE · 2022-07-03

**Rating:** 4
**Confidence:** 4
**Soundness:** 2 fair
**Presentation:** 2 fair
**Contribution:** 2 fair

**Summary:**

This paper presents a generalizable and sample-efficient algorithm for multi-agent coordination in decentralized general-sum games without any access to other agents’ rewards or observations.  It first learns the distributions over the return of individuals and estimates a dynamic risk-seeking bonus to encourage agents to discover risky coordination strategies. Then it proposes an auxiliary opponent modeling task so that agents can infer their opponents’ type and dynamically alter corresponding strategies during execution. Empirical studies show that the trained agents outperform other baseline methods via achieving mutual coordination during training and avoiding being exploited by non-cooperative opponents during execution.

**Questions:**

After reading the paper, I have some questions:

1. Sec. 4.2 is not clear. A figure showing the architecture is needed.

2. Why left truncated variance is used here? Why did the index start from M/2?

3. Why c_tj is used here?  Why combing Eqn. 4 and Eqn. 5 in Eqn. 6 is reasonable? It lacks the motivation

4. LIAM is actually a single-agent RL method, how did you conduct the experiments in MARL scenarios?

5. Did you compare your method with LOLA [3]?

[3] Learning with Opponent-Learning Awareness


**Limitations:**

Please see the above comments.

**Strengths And Weaknesses:**

Strengths: Risk-sensitive learning in zero-sum games is important topic in multi-agent systems. This paper applies risk-sensitive learning in zero-sum games. It uses the well-received distributional RL to learn return distributions and uses distribution distortion. The opponent learning was used to avoid overfitting opponents’ coordination strategies during training.

Weaknesses:
1. This paper combines many well-studies methods and applies them in zero-sum game scenario. The methodological contributions are limited.

2. The evaluation scenarios are mainly on 2-agent cases, which is simple for didactic studies. However, it is hard to see if the proposed method can also perform well on complex 3+ agent scenarios. For example, social dilemma and zero-sum games scenarios in melting pot [1].

3. The writing is not good. Sec. 4.2 is not clear. It is not easy to follow. A figure showing the architecture is needed.

4. The Monster-hunt and Escalation scenarios were modified in [2], which is not cited in this paper.

[1] Scalable Evaluation of Multi-Agent Reinforcement Learning with Melting Pot

[2] DISCOVERING DIVERSE MULTI-AGENT STRATEGIC BEHAVIOR VIA REWARD RANDOMIZATION

---

> ### Author Response · Authors · 2022-07-28
> **Responses to Reviewer metE**
>
> We thank the reviewer’s valuable feedback on our work. We summarize the reviewer’s questions and present our responses below.
> ***
> Q1:The Monster-hunt and Escalation scenarios were modified in [2], which is not cited in this paper.
>
> A1: This is a misunderstanding. We had cited these two scenarios in the last paragraph of Sec. 1, line 63 in the original version.
> ***
> Q2: Sec. 4.2 is not clear. A figure showing the architecture is needed.
>
> A2: Thanks for your advice. We will plot the architecture figure and you can find it in our paper of the rebuttal revision.
> ***
> Q3: Why left truncated variance is used here? Why did the index start from M/2?
>
> A3: Thanks for your valuable question. (1) A naive approach to exploration would be to use the variance of the estimated distribution as a bonus. As shown in [C], the exploration bonus from truncated variance outperforms bonus from the variance. The Right Truncated Variance tells about lower tail variability and the Left Truncated Variance tells about upper tail variability. For instantiating optimism in the face of uncertainty, the upper tail variability is more relevant than the lower tail, especially if the estimated distribution is asymmetric. Intuitively speaking, $\sigma_{+}^{2}$ is more optimistic. $\sigma_{+}^{2}$ is biased towards positive rewards. To increase stability, we use the left truncated measure of the variability, $\sigma_{+}^{2}$.
>
> (2) The index starts from the median, i.e., M/2, rather than the mean due to its well-known statistical robustness [A, B, C].
>
> References:
>
> [A]Huber, Peter J. "Robust statistics." International encyclopedia of statistical science. Springer, Berlin, Heidelberg, 2011. 1248-1251.
>
> [B]Rousseeuw, Peter J., et al. Robust statistics: the approach based on influence functions. John Wiley & Sons, 2011.
>
> [C]Mavrin, Borislav, et al. "Distributional reinforcement learning for efficient exploration." International conference on machine learning. PMLR, 2019.
> ***
> Q4: Why c_tj is used here? Why combing Eqn. 4 and Eqn. 5 in Eqn. 6 is reasonable? It lacks the motivation
>
> A4: Thanks for your valuable question. (1) As shown in [A], the estimated QR distribution is a mixture of parametric and intrinsic uncertainties. As learning progresses the parametric uncertainty vanishes and the intrinsic uncertainty stays. Therefore, this left truncated variance exploration bonus will tend to be biased towards intrinsic variation, which hurts performance. To suppress intrinsic uncertainty, we need a decaying schedule. From the classical QR theory [B], it is known that parametric uncertainty decays at the following rate:
> $$
> c_t=c\sqrt{\frac{\log t}{t}}
> $$
> Where c is a constant factor. So we use c_tj as the decaying schedule.
>
> (2) The left truncated variance defined in Eqn.4 enhances the agent's exploration ability and makes the agent optimistic in the face of uncertainty, and the risk-seeking exploration bonus defined in Eqn.6 encourages agents to select actions that have higher potential payoffs. The ablation study in Sec 5.5 shows that these two objectives are equally important for agents to achieve coordination strategies efficiently.
>
> References:
>
> [A]Mavrin, Borislav, et al. "Distributional reinforcement learning for efficient exploration." International conference on machine learning. PMLR, 2019.
>
> [B]Koenker, Roger, and Kevin F. Hallock. "Quantile regression." Journal of economic perspectives 15.4 (2001): 143-156.
> ***
> Q5: LIAM is actually a single-agent RL method, how did you conduct the experiments in MARL scenarios?
>
> A5: Thanks for your valuable question. (1) In my opinion, LIAM can be viewed as either a multi-agent RL method that focuses on opponent modeling or a single-agent RL method that tackles non-stationary problems. However, in the LIAM paper, paper authors evaluate LIAM in multi-agent scenarios and assume other agents have fixed policies. They use recurrent auto-encoder to model the relationship between the trajectory of the controlled agent and the modeled agents.
>
> (2) In our experiment, each agent is the controlled agent and equipped with the LIAM method to model opponents. In other words, we do not assume opponents have pre-trained fixed policies.
> ***
> Q6:Did you compare your method with LOLA [3]?
>
> A6:  Thanks for your valuable question. We didn't compare our method with LOLA in the original version because LOLA has too strong assumptions that it can access to opponents' parameters and gradient information. And we focus on studying multiple independent learning agents, so that is an unfair comparison. However, according to your suggestion, we will supplement the comparison experiment in the rebuttal revision.

---

> > ### Comment · Reviewer_DtFh · 2022-08-06
> > **Comments on comments.**
> >
> > 1. I'd also *love* to see a comparison to melting pot or advanced social dilemmas. Think the problem is these environments are just not computationally tractable unless the authors sit at a major commercial lab. I think the experiments on multi-step matrix games are the best you can do on an academic computational budget atm.
> >
> > 2. I agree the methodological contributions seem limited but the outcome - independent RL agents cooperating in general sum games is a very exciting one!
> >
> > 3. Kinda agree a comparison with LOLA is a bit unnecessary here (although kudos to the authors for getting that out). LOLA has tonnes of privilege information about an opponent (access to their weights) and its variants which don't, perform strictly worse. The only variant of opponent-shaping with similar limited access (no access to opponents policy) is MFOS [1]. This method is not currently scalable so unclear if it can be applied in the test environments here.
> >
> > [1]  https://arxiv.org/pdf/2205.01447.pdf

---

> > > ### Author Response · Authors · 2022-08-07
> > > **Thank You**
> > >
> > > Dear Reviewer DtFh,
> > >
> > > We are glad to hear that you agree the outcome is a very exciting one. Thank you very much for your feedback.
> > >
> > > Sincerely yours,
> > >
> > > Authors of Paper2297

---

> ### Author Response · Authors · 2022-08-07
> **Dear Reviewer metE, did our responses address your questions?**
>
> Dear Reviewer metE,
>
> As the response system will be closed soon within a few days. We thank you again for your comments. We hope our detailed responses could address your questions. More questions on our paper are always welcomed! If there are no more questions and we will appreciate it if you can kindly raise the score.
>
> Sincerely yours,
>
> Authors of Paper2297

---

### Official Review · Reviewer_ZHUo · 2022-07-09

**Rating:** 3
**Confidence:** 3
**Soundness:** 2 fair
**Presentation:** 2 fair
**Contribution:** 2 fair

**Summary:**

This paper considers the problem of coordination in the more realistic general-sum game settings, by applying techniques in distributional reinforcement learning which allows the agents to leverage risk measures to guide the exploration toward risky cooperative strategies. Besides, an auxiliary opponent modeling task is used to improve the generalizability in face of non-cooperative players.
The method is compared with previous works in four tasks, and the empirical results demonstrate that the proposed method could improve the coordination ability of agents even without access to other agents' rewards, and quickly adapt to non-cooperative agents by fine-tuning the feature extractor.

**Questions:**

1. Since long-term fine tuning is needed at test phase to achieve generalization, why keep $\psi^i_a$ fixed and not tuning anymore?
2. In line 285, you claim that the cooperative agents and non-cooperative agents are trained by GRSP and MADDPG, respectively. Can you explain why these two different algorithms can produce these two desired strategies?
3. The proposed method seems to work well on the 5 by 5 grid-world environments, which left me wondering how well it performs on other popular Markov game environments. For example, social dilemma extends the prisoner's dilemma into multi-step settings, and the results on it can be more convincing.
4. Can you explain why your method can converge to the highest global reward in IPD. This is very unintuitive because it is not an equilibrium, implying that self-interest agents would deviate from it and try to defect at last. Your method does not take the rewards of other agents into consideration and the learned policy is self-interest even though you have incorporated risk-seeking bonus.

**Limitations:**

1. Experiments in this work consists of "Comparison of Returns" only. More comprehensive demonstration experiments can better explain why the method works.
2. Generalization of the method used in this paper requires long-term fine tuning, while a lot of works can tackle ad-hoc teamwork efficiently right now. The application value of this work is limited.
3. During test time, the adaptation needs access to the full action history of all other agents, and fine-tune the feature extractor on it. This can be a very restrictive assumption and limits its use in practice.

**Strengths And Weaknesses:**

Strengths:

1. This paper takes a look into a valuable point. Coordination is often studied in the fully cooperative setting, but general-sum game is more realistic. This paper considers an important but less studied problem. Also, generalization is a major but often overlooked issue in current MARL methods.
2. The proposed method applies the risk-seeking bonus in distributional RL to solve the exploration problem in multi-agent tasks, which is well-motivated and has the potential to improve the coordination ability.
3. The method used in this article combines risk-sensitive learning and agent modeling, both of which are promising techniques to handle the non-stationarity caused by environmental and agent level changes in multi-agent settings.

Weaknesses:

1. The most important part method is too short compared with the preliminaries and experiments. It is unclear to me how the feature extractor $\mathcal{E}$ and the decision maker $D$ work, more details are needed to introduce how your framework functions as a whole.
2. 1. There have been some works considering risk-sensitive or distributional Q-functions in MARL, such as [DFAC](https://arxiv.org/abs/2102.07936) and [D4PG](https://ieeexplore.ieee.org/document/9311945). This part of the method in this paper seems to be an implementation of risk-sensitive Q-function upon independent Q-learning (IQL), thus having rather restricted innovations.
3. The risk-seeking bonus results in optimistic behavior during training, this is beneficial when training with cooperative agents, but on the other side, which can also be exploited by non-cooperative agents and thus fail to learn a generalizable cooperative policy.
4. The paper claims to have generalization ability across different opponents during execution, but the way of achieving this requires long-term fine tuning during test phase (even just a part of the network will be tuned). I think such level of generalization should be expressed as "policy transferring" for clarity in order not to confuse others to think of zero-shot coordination (ZSC) or ad-hoc teamwork (AHT).

Minors:
1. The equation 7 makes it hard to understand what is the output of the function $D$. Besides, the parameters to be optimized on the LHS are $\phi^i$ and $\psi^i_s$, but $\phi^i$ disappears on the RHS.
2. The loss term $\mathcal{J}$ in equation 8 is not defined. I know that you have introduced it in the preliminaries, but please define it again under your framework according to your parameterization.

---

> ### Author Response · Authors · 2022-07-29
> **Responses to Reviewer ZHUo**
>
> We thank the reviewer’s valuable feedback on our work, and we present our responses below.
>
> Weakness1 and Minors 1, 2: We have revised our paper according to your valuable suggestions and fixed typos in equation 7, and the rebuttal revision has been updated.
>
> Weakness2 and 3: Both D4PG and DFAC focus on fully cooperative setting and do not consider risk-sensitive policies. To our knowledge, our work firstly proposed the algorithm which utilizes risk-seeking policies to achieve risky coordination strategies in general-sum games that is a very general problem, and firstly studied how the lower tail of the quantile distribution affects action selection and corresponding payoffs. We further proposed a novel opponent modeling algorithm to generalize the agent's policy to different opponents, thus avoiding being exploited by non-cooperative opponents during testing time. And we achieve the best and strongest results in four widely used experimental environments with the weakest assumptions.
>
> Weakness4: In fact, agents only need several episodes (less than ten) to adapt to non-cooperative opponents. You are welcome to download our codes and models in https://anonymous.4open.science/r/GRSP-8DEC/README.md to test them.
> ***
> A1: Recall that $\phi^{i}$ are parameters of the feature extractor $\mathcal{E}_{\phi^{i}}$
>
> and $\psi_{a}^{i}$ are parameters of the decision maker
> $\mathcal{D}_{\psi_{a}}$.
>
> During training, $\phi^{i}$ can be updated by gradients from both RL objective and auxiliary opponent modeling task, and  $\psi_{a}^{i}$ can be updated by gradients only from RL objective.
>
> During testing, agents can not access the environmental rewards, so $\phi^{i}$  can't be tuned anymore through RL objective. Only the feature extractor $\mathcal{E}_{\phi^{i}}$ can be tuned by gradients from the auxiliary opponent modeling task which only needs history actions made by the opponent.
> ***
> A2: In our experiments,  GRSP is the only method that can produce cooperative strategies. So we use GRSP to train cooperative agents. All other baseline methods can only produce non-cooperative agents in our experiments,  and since MADDPG is a popular and easy-to-implement algorithm, we choose it to produce non-cooperative agents.
> ***
> A3: Our prisoner's dilemma environment is naturally a multi-step Markov game and is widely used in [A,B,C,D]. Furthermore, the experimental results of our method are the best among others, e.g., the superior performance and no extra assumptions. All learning agents can achieve coordination strategies with our method and only need the risk-seeking bonus. This is the best result in the sequential social dilemma so far.
>
> [A] Foerster, Jakob N., et al. "Learning with opponent-learning awareness." arXiv preprint arXiv:1709.04326 (2017).
>
> [B] Tang, Zhenggang, et al. "Discovering diverse multi-agent strategic behavior via reward randomization." arXiv preprint arXiv:2103.04564 (2021).
>
> [C] Leibo, Joel Z., et al. "Multi-agent reinforcement learning in sequential social dilemmas." arXiv preprint arXiv:1702.03037 (2017).
>
> [D]Wang, Weixun, et al. "Towards cooperation in sequential prisoner's dilemmas: a deep multiagent reinforcement learning approach." arXiv preprint arXiv:1803.00162 (2018).
> ***
> A4: The Iterated Prisoner's Dilemma(IPD) is a kind of Repeated Game. The equilibria of a Repeated Game can differ from those of the associated strategic game. In the strategic game prisoner's dilemma, the only Nash equilibrium is defection-defection. However, in IPD, the strategy that chooses to cooperate after every history is the best response to the strategy that starts off choosing to cooperate and “punishes” any defection by switching to defect. If one agent chooses to defect, then the other agent will alter its strategy from cooperation to defection in the next round and both of them will get lower payoffs in the end. In infinitely IPD, if the discount factor $\delta \geq 0.5$, then cooperation is the best response to cooperation[A]. We quote [B] to further explain this:
>
> >  in the infinitely repeated game, there are an infinite number of equilibria. So equilibrium selection becomes a problem. But cooperative strategies generally form equilibria with each other: if you are playing grim-trigger, and I am playing tit-for-tat, and we both sufficiently value the possibility of future play, then we will keep cooperating and neither of us can do any better with a different strategy.
>
> In our method, the risk-seeking exploration bonus will encourage agents to choose actions that have much higher potential cumulative rewards, i.e., cooperation. And the auxiliary opponent modeling task can alter the agent's strategy from cooperation to defection if its opponents choose to defect. The two components can constitute equilibrium strategies between agents.
>
> [A] https://economics.mit.edu/files/4754
>
> [B] Levin, N. (Ed). (2019). Introduction to Ethics: An Open Educational Resource. N.G.E. Far Press.

---

> ### Author Response · Authors · 2022-08-07
> **Dear Reviewer ZHUo, did our responses address your questions?**
>
> Dear ReviewerZHUo,
>
> As the response system will be closed soon within a few days. We thank you again for your comments. We hope our detailed responses could address your questions. More questions on our paper are always welcomed! If there are no more questions and we will appreciate it if you can kindly raise the score.
>
> Sincerely yours,
>
> Authors of Paper2297

---

> > ### Comment · Reviewer_ZHUo · 2022-08-07
> > **Response to Author Rebuttal**
> >
> > Thanks for the clarification, but the question 4 is still unclear. This is related to the generality of the method as the author didn’t put any restrictions on the range of problems it can solve, so I will still maintain my current score.
> >
> > My question actually echoes the major concern of reviewer DkS2, which is under what condition can the proposed algorithm converge to the risky cooperative policies. In the tasks ISH, Monster-Hunt, and Escalation, where the optimal social welfare and optimal individual value can be achieved at the same time, it is reasonable for your method to converge to the cooperative strategies.
> >
> > However, IPD is a totally different story, where agents can always be better off by choosing to defect and this is why it’s called a dilemma. Given that the the policy optimizes the episodic individual reward and the coefficients of the risk bonus are decreasing, agents will definitely deviate from the cooperative policies if trained long enough. The author also mentioned the opponent modeling task, but it doesn’t make any change to the objective of the policy. Can the opponent modeling task help to achieve coordination during test? Maybe, but effect of the modeling task is only on the hidden layers of the policy, it cannot fully explain why it helps.

---

> > > ### Author Response · Authors · 2022-08-07
> > > **Responses to Reviewer ZHUo**
> > >
> > > We thank the reviewer's feedback. We answer your question "under what condition the proposed algorithm converges to the risky cooperative policies" as follows:
> > >
> > > Perhaps you have a misunderstanding of our method and repeated prisoner's dilemma (or iterated prisoner's dilemma). IPD is a multi-stage markov game, and PD is a one-step matrix game. Agents can be better off by choosing to defect in PD. However, this is not true in IPD when all agents can update their strategies. Consider two players play 30 steps IPD, they begin with cooperative strategies, and all agents get 2 rewards at each time step. Then at time step $11$, agent 2 chooses to defect and keeps defect from now on. It gets 3 rewards at time step $11$. However, agent 1 can observe agent 2's choice and alter its strategies. Assume at time step 16, agent 1 alters its strategy from cooperation to defection. Next, all agents get 0 rewards until the episode ends. So the rewards of agent 1 is 10 * 2 -1 * 5=15, and the rewards of agent 2 is 10 * 2 + 3 * 5=35. However, if all agents keep cooperation until the episode ends, each of them will get 50 * 2=100 rewards. So in the long run, a cooperative strategy can make agents achieve both optimal social welfare and optimal individual value at the same time. No one will deviate from cooperation to defection because their partners will defect, too. This equilibrium has been proved in the game theory community [1][2][3][4].
> > >
> > > Although the risk bonus is decreasing, agents have accumulated ample high return samples, i.e., actions leading to cooperation. These samples will change the return distribution shape, and agents' exploration bonus are decreasing, too. So in the end, the action which has the max Q value is the cooperation action, not the defection action.
> > >
> > > The policy is trying to maximize individual value, so we didn't change the objective of the policy in the general-sum game. Maximizing individual value is the most important thing in the general-sum game. Defect can bring equal or higher payoff in the short term, but it is not in the long run, including ISH, Monster-Hunt, Escalation and, IPD. Cooperation will make all agents get the max individual value and the optimal social welfare. But cooperation has risks. Learning agents with our method can stand this risk for better future returns. So if all agents equip with our GRSP method, they will achieve cooperation in the end.
> > >
> > > **As long as all agents choosing cooperative policies can make everyone get better payoffs in the long run, agents with our method can converge to the risky cooperative policies.**
> > >
> > > As for the auxiliary opponent modeling( AOM) task, our opponent modeling objective and policy objective are **optimized jointly**. If we have a cooperative opponent, the AOM objective will update policy's parameters to get a better payoff. So the AOM objective will force the policy to choose cooperation. If we have a defective opponent, the AOM objective will update the policy to change its behaviors from cooperation to defection, avoiding being exploited.
> > >
> > > ***
> > > [1] Fudenberg, Drew, and Tirole, Jean. (1991) Game Theory. Cambridge, MA. The MIT Press.
> > >
> > > [2]Myerson, Roger B. (1991) Game Theory: Analysis of Conflict. Cambridge, MA. Harvard University Press.
> > >
> > > [3]Poundstone, William. (1992) Prisoner’s Dilemma. New York, NY. Doubleday.
> > >
> > > [4]Axelrod, Robert M. (1984) The Evolution of Cooperation. New York, NY. Basic Books, Inc.

---

> > > > ### Comment · Reviewer_ZHUo · 2022-08-09
> > > > **Thanks. I still have some questions.**
> > > >
> > > > Thanks for your detailed response. I still have some questions about the tasks that the paper can solve. The paper may be able to solve IPDs since there's an optimal solution for both agents to choose "cooperate" when the horizon is limited, or there is a discounting factor. The paper distinguishes actions in multi-agent environments as "risk-seeking" and "risk-avoiding" and combines them with "cooperate" and "defect" in Prisoner's Dilemma. However, real multi-agent scenarios are more complex and may not have the basic form of IPDS. There exist plenty of environment settings where action space cannot be divided into these two categories, or this correspondence does not exist. For example, how can you define "cooperate" and "defect" actions in the symmetric matrix games below and apply your risk-seeking method? I don't think evaluating risk in such an environment can be effective since the payoff for each action is equal without knowing the teammate. And if your agent develops some bias of actions during training (e.g., prefer to choose $(a_1,a_1)$), it will fail to coordinate with new teammates that prefer to do the other action ($a_2$).
> > > >
> > > >    |           |  P2: $a_1$   |  P2: $a_2$   |
> > > >    | :-------: | :----------: | :----------: |
> > > >    | P1: $a_1$ |  $(10, 10)$  | $(-10, -10)$ |
> > > >    | P1: $a_2$ | $(-10, -10)$ |  $(10, 10)$  |

---

> > > > > ### Author Response · Authors · 2022-08-09
> > > > > **Responses to Reviewer ZHUo**
> > > > >
> > > > > We thank reviewer ZHUo's valuable feedback and your efforts in the whole rebuttal stage. We answer your questions as follows:
> > > > >
> > > > > 1. The risk in our paper characterized the length of the quantile distribution's lower tail, which means the potential loss of the agent's action choice. Why do we need a risk-seeking bonus? Because the action which has high risk, i.e., high potential loss, also has a higher potential payoff at the same time. So we can adjust WT's hyperparameter $\lambda$ to construct a risk-seeking bonus to encourage agents to choose high-risk but higher payoff actions.
> > > > >
> > > > > 2. We can also adjust WT's hyperparameter $\lambda$ to 0 to construct a risk-neutral agent in order to tackle an environment that has no risk, e.g., the matrix game you presented. Now the risk-seeking bonus becomes a risk-neutral bonus, i.e., the action expectation of quantile distribution. Our method can construct an agent that is risk-seeking, risk-neutral, or risk-averse, depending on what kind of environmental dynamics the agent interacts with.
> > > > >
> > > > > 3. We conducted a new experiment using the matrix game you presented. With the same hyperparameters as IPD agents, our agents can converge to 100 rewards (keep cooperation in the whole episode, which has 10 steps) during training after 23 episodes. We also conducted an experiment with risk-neutral agents, they can achieve cooperation-cooperation after 22 episodes, too.
> > > > >
> > > > > 4. "And if your agent develops some bias of actions during training (e.g., prefer to choose (a1,a1)), it will fail to coordinate with new teammates that prefer to do the other action (a2).":
> > > > >
> > > > > Your concern is correct. Both single-agent and multi-agent training methods will prefer to overfit their environment or opponents. How to overcome it is a very essential and core problem in reinforcement learning. In the single-agent community, it is the concern of meta-learning and multi-task learning research[1]. In the multi-agent community, it is the concern of zero-shot coordination[2] and ad-hoc teamwork research[3]. Our opponent modeling method offers a new perspective on these communities. We acknowledge that it is not a strong enough method to cover every situation or even end the research in this field. Our opponent modeling method is suitable for situations where there exists a safe equilibrium that each agent gets a lower payoff but can avoid being exploited. As for the matrix game you presented, we can train ensemble cooperation policies to equip each agent and dynamically select which policy to use when coordinating with new teammates.
> > > > >
> > > > > We hope our answer can eliminate your confusion.
> > > > >
> > > > > [1] Rakelly, Kate, et al. "Efficient off-policy meta-reinforcement learning via probabilistic context variables." International conference on machine learning. PMLR, 2019.
> > > > >
> > > > > [2] Lupu, Andrei, et al. "Trajectory diversity for zero-shot coordination." International Conference on Machine Learning. PMLR, 2021.
> > > > >
> > > > > [3] Mirsky, Reuth, et al. "A Survey of Ad Hoc Teamwork: Definitions, Methods, and Open Problems." arXiv preprint arXiv:2202.10450 (2022).

---

### Official Review · Reviewer_DtFh · 2022-07-09

**Rating:** 9
**Confidence:** 4
**Soundness:** 4 excellent
**Presentation:** 4 excellent
**Contribution:** 4 excellent

**Summary:**

The paper introduces GRSP (Generalizable Risk-Sensitive MARL algorithm), a method for training agents in mixed-games to converge to cooperative nash-equilibrium. The method's impact is in its applicability to much more relaxed situations than other MARL algorithms (usually Centralised Training Decentralised Execution). The key insights for this method are two fold:

1) cooperative equilibrium are usually “high-risk” so simple reward maximisation rarely converges to this.

2) instead of requiring full access to co-players during training, simply being provided their action is suffice for opponent modelling. This is closer to independent learning than existing methods.

The method is then applied in two evaluation protocols: Evaluation Of Returns during training (can they converge to the desired equilibrium) and Generalisation to Novel Co-players at test-time. In both situations compelling arguments are made to show GRSP efficacy. Finally ablations are applied to distinguish how important the “high-risk” vs “opponent-modelling” insights and associated components are.


**Questions:**


I’d like more understanding of the training dynamic:


1. How does using the distribution RL effect sample efficiency?
2. How does risk-sensitive approaches effect exploration?



**Limitations:**

None to mention :)


**Strengths And Weaknesses:**


1.  Clean presentation
2.  Good experiment protocol
3. Very good breath of environments
4. Solid theoretical justification
5. Original work.
6. Clearly explains intuitions for method.


Weaknesses;
1) Very little time is given to explain training models, i’m unsure how sample efficient this method is compared to others.
2) Generalisation Study Results could be better displayed. The opponent shaping literature has much nicer payoff graphs for showing how much an agent has cooperated with another -> these could help explain your findings.

3) I have little understanding of the distortion methods discussed would be useful.

---

> ### Author Response · Authors · 2022-07-31
> **Responses to Reviewer DtFh**
>
> We would like to thank you for your review. We are delighted that you found the paper well-written and well-executed. We summarize your questions and present our responses below.
> ***
> Q1: How does using the distribution RL effect sample efficiency?
>
> A1: As shown in the ablations part Sec. 5.5 in the original version or Sec 5.4 in the rebuttal revision,  IQR-DQN performs poorly and does not show sample efficiency compared with GRSP and other baseline methods, which indicates that the sample efficiency of GRSP comes from our risk-seeking exploration bonus and truncated variance, and our ablation experiments further demonstrate it empirically. As for distributional RL itself, [A] summarizes that possible reasons for DRL's superiority include the following:
>
> 1. Reduced chattering: modeling a distribution may reduce prediction variance, which may help in policy iteration.
> 2. Improved optimization behaviour: distributions may present a more stable learning target, or in some cases(e.g. the softmax distribution used in the C51 algorithm) have a regularizing effect in optimization for neural networks.
> 3. Auxiliary tasks: the distribution offers a richer set of predictions for learning, serving as a set of auxiliary tasks which is tightly coupled to the reward.
>
> We hope we answered your questions.
>
> [A] Lyle, Clare, Marc G. Bellemare, and Pablo Samuel Castro. "A comparative analysis of expected and distributional reinforcement learning." Proceedings of the AAAI Conference on Artificial Intelligence. Vol. 33. No. 01. 2019.
> ***
> Q2: How does risk-sensitive approaches effect exploration?
>
> A2: Let's first introduce two kinds of uncertainty in RL: epistemic uncertainty, which stems from limited data, and aleatoric uncertainty, caused by intrinsic stochasticity in the environment, and the epistemic uncertainty will vanish as the learning progress. In MARL, we claim that the aleatoric uncertainty, i.e., the intrinsic stochasticity in the environment, is caused by other learning agents' exploration and policy updating. Distributional RL, which aims to learn the distribution of returns instead of the mean value only, has been suggested as a way of measuring aleatoric uncertainty[A]. So we adopted distributional RL to capture the aleatoric uncertainty in general-sum games.
>
> Furthermore, the action whose value distribution has a longer upper tail means that taking this action may have higher potential payoffs. However, its mean value may be lower than other actions since its distribution also has a longer lower tail, which means higher risk. So agents with the expected RL method will not select this action. We proposed to utilize the risk-seeking exploration bonus to encourage agents to pay more attention to actions whose distribution has a longer upper tail. So agents with our GRSP method will be less likely to defect to others since defects bring lower future returns, and more likely to coordinate with each other although coordination has a bit of risk.
>
> In a word, the risk-seeking approach will encourage agents to explore regions that have higher potential future returns and be more tolerant of the risk.
>
> We hope we answered your questions.
>
> [A] Nikolov, Nikolay, et al. "Information-directed exploration for deep reinforcement learning." arXiv preprint arXiv:1812.07544 (2018).

---

### Author Response · Authors · 2022-08-06
**Dear reviewers, did our responses address your questions?**

We thank again all the reviewers for their constructive and valuable comments. We really appreciate the positive comments made by reviewers who recognised our contribution to MARL.

We hope our responses, including extra experiments and revised paper, could address the questions of all the reviewers. More discussions and suggestions on our paper are also always welcomed!

Sincerely yours,

Authors of Paper2297

---

### Meta-Review · Area_Chair_FpVN · 2022-08-25

**Recommendation:** Reject
**Confidence:** Less certain

**Metareview:**

The paper presents a novel approach for improving coordination in general-sum games by using risk-sensitive policies based on distributional RL. While the idea is promising, there are significant questions about the paper.

For example, there is concern about the lack of theoretical guarantees and intuition about when the approach will work well.

There should also be a more extensive discussion of related work. For example, distributional and risk-sensitive RL have been used in multi-agent RL but more should be said about how the proposed method differs and why they can't be included in the experiments (e.g., the decentralized training methods should have no problem running in the general sum case).

More extensive experiments would also be helpful. Additional domains and baselines would more clearly show the benefits of the method (and when it could potentially fail).

**Award:**

No

---

### Decision · Program_Chairs · 2022-09-14

Reject